# Comparative Whole Genome Analysis of an *Anaplasma phagocytophilum* Strain Isolated from Norwegian Sheep

**DOI:** 10.3390/pathogens11050601

**Published:** 2022-05-21

**Authors:** Francy L. Crosby, Sveinung Eskeland, Erik G. Bø-Granquist, Ulrike G. Munderloh, Lisa D. Price, Basima Al-Khedery, Snorre Stuen, Anthony F. Barbet

**Affiliations:** 1Department of Infectious Diseases and Immunology, College of Veterinary Medicine, University of Florida, Gainesville, FL 32608, USA; balkhedery@yahoo.com (B.A.-K.); barbet@ufl.edu (A.F.B.); 2Department of Production Animal Clinical Sciences, Section of Small Ruminant Research, School of Veterinary Medicine, Norwegian University of Life Sciences, Oslo 1432, Norway; sveinung.eskeland@nmbu.no (S.E.); erikgeorg.granquist@nmbu.no (E.G.B.-G.); snorre.stuen@nmbu.no (S.S.); 3Department of Entomology, College of Food, Agricultural and Natural Resources, University of Minnesota, St. Paul, MN 55108, USA; munde001@umn.edu (U.G.M.); pric0129@umn.edu (L.D.P.)

**Keywords:** *Anaplasma phagocytophilum*, comparative genomics, co-infection, lateral gene transfer, ticks, tick-borne fever

## Abstract

*Anaplasma phagocytophilum* is a Gram-negative obligate intracellular tick-borne alphaproteobacteria (family Anaplasmatacea, order Rickettsiales) with a worldwide distribution. In Norway, tick borne fever (TBF), caused by *A. phagocytophilum*, presents a major challenge in sheep farming. Despite the abundance of its tick vector, *Ixodes ricinus*, and *A. phagocytophilum* infections in wild and domestic animals, reports of infections in humans are low compared with cases in the U.S. Although *A. phagocytophilum* is genetically diverse and complex infections (co-infection and superinfection) in ruminants and other animals are common, the underlying genetic basis of intra-species interactions and host-specificity remains unexplored. Here, we performed whole genome comparative analysis of a newly cultured Norwegian *A. phagocytophilum* isolate from sheep (ApSheep_NorV1) with 27 other *A. phagocytophilum* genome sequences derived from human and animal infections worldwide. Although the compared strains are syntenic, there is remarkable genetic diversity between different genomic loci including the pfam01617 superfamily that encodes the major, neutralization-sensitive, surface antigen Msp2/p44. Blast comparisons between the *msp2/p44* pseudogene repertoires from all the strains showed high divergence between U. S. and European strains and even between two Norwegian strains. Based on these comparisons, we concluded that in ruminants, complex infections can be attributed to infection with strains that differ in their *msp2/p44* repertoires, which has important implications for pathogen evolution and vaccine development. We also present evidence for integration of rickettsial DNA into the genome of ISE6 tick cells.

## 1. Introduction

*Anaplasma phagocytphilum* is a bacterial pathogen that has long been known to cause tick-borne fever in sheep. Recently, it has also been identified as the causative agent of an emerging disease in humans, known as human granulocytic anaplasmosis (HGA), with > 5000 cases identified in the U.S. by the CDC in 2019 (https://www.cdc.gov/anaplasmosis/stats/index.html accessed on 24 March 2022). In Norway, an estimated ~300,000 sheep are infected annually, with severe economic and animal welfare consequences [1,2]. This has stimulated the search for a vaccine against the disease in sheep. Although continuous culture of different geographic strains of *A. phagocytophilum* has been achieved, these overwhelmingly represent U.S. strains derived from humans isolated in the human promyelocyte cell line HL-60. Culture of sheep *A. phagocytophilum* strains has proven more difficult because they do not grow in HL-60 cells. Nevertheless, ruminant strains of *A. phagocytophilum* can be isolated using tick cell lines [3,4,5], and we recently obtained cultures of two strains from experimentally-infected Norwegian sheep, known as ApSheep_NorV1 and ApSheep_NorV2, in the tick cell line ISE6 [2]. Of these two strains, ApSheep_NorV1 is of particular interest for vaccine development because of its virulence [6]. However, it was recently shown that ApSheep_NorV1, derived from ISE6 cells, was not as immunoprotective in lambs as the original field strain derived from infected sheep blood [2]. Here, we identify a possible reason for this discrepancy, and provide a genomic analysis of the cultured ApSheep_NorV1 strain in comparison with other *A. phagocytophilum* strains of diverse animal and geographic origins. 

## 2. Results and Discussion

### 2.1. Characterization of the ApSheep_NorV1 Strain of A. phagocytophilum

In the U.S., small nucleotide changes within a variable region near the 5’ end of the 16S rRNA gene have been used to identify *A. phagocytophilum* variants that appear to either only infect humans (Ap-ha) or ruminants (Ap variant 1) [7,8]. In Europe, *A. phagocytophilum* 16S rDNA variants identified in ruminants are related to the Ap-ha and Ap variant 1 found in the U.S. However, contrary to the host tropism observed for these variants in the U.S., in Europe, both Ap-ha and Ap variant 1 have been detected in ruminants. Based on this variability in the 16S rDNA, several variants have been identified in sheep in Norway. Among these, ApSheep_NorV1 is the most virulent, and possesses partial 16S rDNA sequence identity to the sequence with accession number M73220 [9]. Previous attempts to assemble a complete genome for this variant using 454 sequencing reads from bacteria derived directly from infected blood collected during acute rickettsemia were unsuccessful. Instead, this analysis revealed heterozygous copies of single-copy genes that, given the haploid nature of the *A. phagocytophilum* genome, evidenced a mixed infection (Figure 1). To estimate the proportions of each population, we determined the number of reads that align to sites known to vary among *A. phagocytophilum* strains, i.e., sequencing reads that bind to a variable region of *ompX* and *virB6-2* (Table 1). This analysis indicated the presence of a major and a minor genetic population represented by 89% and 11% of the reads, respectively. To confirm this analysis, a PCR with primers targeting the *virB6-2* gene to differentiate between the major and minor variants was performed using genomic DNA extracted from blood samples collected at consecutive rickettsemia peaks from two lambs infected with ApSheep_NorV1; animal #4203 (5 rickettsemia peaks) and animal #4210 (4 rickettsemia peaks) [10,11] (Figure 2). This analysis detected both the major and minor variants during the first acute rickettsemia peak, 8 days post-infection (p.i.), in both animals, confirming that field strain ApSheep_NorV1 is a mixed infection of two different genotypes. However, in lamb #4203 only the minor variant was detected at 22 days p.i. while the major variant only was detected at 50 days p.i. In lamb #4210, only the minor variant was detected again, at 48 days p.i. (Figure 2). Regardless of the lower sensitivity of this endpoint PCR, relative to the real-time PCR performed [10], it is possible that the presence of these major and minor variants varies during the course of infection. It has been shown that infection with more than one *A. phagocytophilum* strain can occur during the acute or persistent phase of infection and compete against each other within the host [12,13]. Additionally, many different 16S genotypes of *A. phagocytophilum* may exist in single sheep flocks and even within infections of single animals [14]. However, 16S rDNA does not reliably identify *A. phagocytophilum* genotypes, and cannot rule out the presence of other variants.

For transposon mutagenesis using the Himar 1 system, we sought to propagate these variants in cell culture [2]. This resulted in the isolation of three mutant populations (referred to as CL1A2, CL2B5, and CL3D3) in the *Ixodes scapularis* ISE6 tick cell line [2]. PCR analysis to determine whether the isolated Himar1-mutants belonged to the major or minor population showed that mutants CL1A2 and CL2B5 represented only the minor genotype while mutant CL3D3 included both genotypes, suggesting a mixed population (Figure 3). We obtained a complete genome sequence of the CL1A2 mutant (representing the minor genotype in ApSheep_NorV1) using the Pacific Biosciences platform. Bioinformatics analysis indicated that CL1A2 carried two Tn insertions in different regions as well as reads containing the unmutated wild-type locus at each region [2]. This indicated that CL1A2 also represented a mixture of two distinct mutant populations, which would explain the presence of wild-type sequences for both mutated loci. Therefore, during genome assembly, the two transposon-containing loci were removed and the final assembly contains only the unmutated wild-type loci (GenBank accession number CP046639) [2]. Culture isolation of *A. phagocytophilum* in either tick or mammalian cell lines imposes its own selection pressures [15]. Further evidence for this was obtained herein, where a cultured strain of *A. phagocytophilum* was only partially representative of the field strain from which it was derived. Clearly, 16S rDNA sequence alignment is not sufficient to fully define an *A. phagocytophilum* genotype and this may explain the poor immune protection observed previously by cultured compared to the original blood stabilate strain.

### 2.2. Comparative Genome Analysis

Sequences of the 16S rRNA gene and the groESL operon are frequently used to identify *A. phagocytophilum* genetic variants. However, these targets do not have enough resolving power to distinguish between *A. phagocytophilum* strains, or to infer host tropisms, especially of human-infective strains globally. We initially compared two U.S. human strains from New York (NY, HZ2) and Massachusetts (MA, NCH1) with the two newly sequenced Norwegian sheep strains using YASS genomic dot-plots [16] (Figure 4). Comparisons of genomic dotplots showed overall synteny between the different *A. phagocytophilum* genomes, and the presence of a genome region known to contain clusters of partially identical *msp2/p44* pseudogenes used for generation of surface antigen diversity in *A. phagocytophilum* [17]. Interestingly, the dotplots showed a similar degree of diversity between the two Norwegian sheep strains and that found between either of these sheep strains and the human strains. This greater conservation of *msp2* repertoires between U.S. human strains is also suggested by the genomic dotplot comparison of the human HZ2 strain from New York state with the NCH1 human strain from Massachusetts (Figure 4). 

We used average nucleotide identity (ANI) between genomes to further investigate genomic relationships between strains of *A. phagocytophilum*. Previous analyses of genome databases using JSpecies software [18] led to the proposal that the boundary between species could be set at ~95–96% ANI. Applying this measure to 28 *A. phagocytophilum* strains resulted in a maximum ANI of 99.99% and a minimum ANI between strains of 94.54%. Human, dog, and rodent strains from the Northeast and Midwestern U.S. were most closely related with an ANI of >99.4% (Appendix A). This level of overall sequence identity was not apparent between most other genomes except for two clusters of *A. phagocytophilum* strains representing either U.S. Ap variant 1 from ticks collected at Camp Ripley, MN (denoted CRT), or a group of bovine strains from France and Germany. The taxonomic relationships between all 28 strains are shown in Figure 5, identifying the clear separation of U.S. from European strains of *A. phagocytophilum* and the clustering of similar groups as observed in the JSpecies analysis [18,19]. These relationships are shown graphically by Blast comparisons between whole genomes (Blast Ring Image Generator, BRIG) [20], using either the ApHZ2_NY genome (innermost circle) or ApSheep_NorV1 genome (outermost circle) as the reference (Figure 6A,B). There is considerable diversity in many genome regions between European strains and even between the two strains infecting Norwegian sheep.

The pan-genome of a given species is composed of the core (genes commonly shared in all strains) and accessory genomes, i.e., genes shared in two or more, but not all the strains (shell genes) and singletons or strain-specific genes (cloud genes) [21]. We performed pan-genome analysis to explore potential differences in gene repertoires among all the 28 *A. phagocytophilum* strains used in this study. This analysis indicated a total of 4561 genes represented by 501 core genes (352 hard + 149 soft core genes) and 4060 accessory genes (1332 shell + 2728 cloud genes). We used the core genome from each *A. phagocytophilum* strain to determine their taxonomic relationship (Figure 7A). In line with whole genome data, core genome analysis also yielded two main groups represented by the U.S. and European strains and, among the U.S. strains, a clear separation between the Ap-variant1 and the Ap-ha strains, respectively. The distribution frequency of accessory genes indicated a cluster of genes found only in the Ap-ha related strains and not in any of the other strains, and a group of genes found only in the Ap-variant-1 or European strains (Figure 7B). 

### 2.3. Comparison of the msp2/p44 Repertoires

In *A. phagocytophilum*, the *msp2/p44* multigene family consists of >100 paralogous genes which are expressed during infection by recombination of the central hypervariable region into a single expression site encoding the major variable surface antigen Msp2/p44 [10,11,15,22]. It has been suggested that these large *msp2/p44* repertoires are important for immunological cross-protection between *A. phagocytophilum* strains [15,22]. Repertoire diversity could be responsible for the lack of protection in sheep immunized with the cultured (minor) genotype of ApSheep_NorV1 when challenged with blood stabilate of the field strain known now to contain two different genotypes [2]. That is, if the antigenic repertoires encoded by the different genotypes are also different, immunization with the cultured strain would be expected to induce an immune response of a narrower specificity that might not protect against challenge with the field strain from which the cultures were derived.

We extracted the repertoires of *msp2/p44* genes from the genome of each strain and determined how many genes were shared at 99% identity or greater in an all-against-all Blastn comparison. We then extended this comparison to include human and animal isolates from the U.S. and Europe. 

There was a surprising amount of diversity between genomic *msp2/p44* repertoires in general (Appendix A), although also a tendency for repertoires of human isolates from Minnesota (MN) to match well with other human isolates from both MN (average percent nucleotide identity ± SD) (98.82% ± 1.6%) and Wisconsin (WI) (98.24% ± 0.83%). From the same general geographic region, the human patient-derived MN repertoires were similar to those from MN rodents (67.06% ± 2.15%), nearly identical to MN dogs (98.82% ± 1.66%), and similar to the repertoires from the WI dog (62.4%). The human-derived WI repertoires were similar to those from WI dog (62.4%) but nearly identical to the MN dog (98.8%). The two human isolates from NY were similar to one another (92.21% ± 5.90%) but divergent from human isolates from MN and WI (50.02% ± 3.50%). The *msp2/p44* repertoires of CRT (Ap variant 1) isolates from MN were not only significantly different to one another (25% ± 3.91%) but also to the repertoires from any of the U.S. human, dog, rodent and horse isolates (7.61 ± 4.75%), suggesting greater evolutionary divergence, possibly as a result of a different vector-host transmission cycle. The observation of just one shared *msp2/p44* gene between a California (CA) and MN horse isolate, and none between European and U.S. horse or dog isolates indicates, however, that the divergence in repertoires is more complex than what would be expected if they simply resulted from adaptation of *A. phagocytophilum* to a particular host species. Indeed, our evidence suggests that *A. phagocytophilum* strains evolve in a regional context, and are isolated from strains circulating in other geographic regions. This effect is seen within regional isolates of both Ap-ha and Ap variant 1 strains. It should be noted that in a previous repertoire analysis [23] that required > 90% identity rather than the value of 99% that we used, 46 *msp2/p44* genes were identified as similar between ApHZ_NY and ApHorse1_CA. This suggests that although there is rapid divergence from identity, some repertoire similarity can be retained. Greater repertoire diversity was apparent between European than between U.S. isolates with the exception of similar repertoires observed in ruminants from the same geographic region in Europe, as was observed previously using overall genomic ANI results. Interestingly, there was little conservation of *msp2/p44* repertoires between the two strains of *A. phagocytophilum* infecting Norwegian sheep, demonstrating repertoire divergence in infections of the same host species from a similar geographic area. It will be interesting to determine how common co-infections are in ruminants, and whether they always involve divergent *msp2/p44* genes. Because Msp2/p44 variants are considered to be cross-protective against variants carrying a similar repertoire, strains expressing antigenically different Msp2/p44 surface proteins should be more likely to co-exist in the same animal than those with a similar repertoire, which would result in immunological competition and exclusion. The taxonomy of *Wolbachia* endosymbionts suggest coevolution with the host given that this organism is strictly vertically transmitted (transovarial transmission from female to offspring) [24], so, it would be possible that divergence of *A. phagocytophilum* strains that infect the same host may, in part, be influenced by the lack of strict vertical transmission. 

*Anaplasma phagocytophilum* is maintained within tick populations by transstadial transmission and/or acquired during bloodmeal from different hosts (horizontal transmission); therefore, ticks may harbor multiple *A. phagocytophilum* strains that differ in virulence and transmissibility. Future studies should also examine *A. phagocytophilum* genomic variation within the tick as there is a lack of knowledge about how *A. phagocytophilum* diversity is maintained and how within-vector competition may drive this pathogen’s evolution and host selection.

These data suggest a possible molecular basis for immunologic cross-protection and superinfection between *A. phagocytophilum* strains employing differential selection from different *msp2/p44* repertoires for recombination into the single expression site. The larger repertoires in *A. phagocytophilum* compared to *Anaplasma marginale* [22,25] may explain why repertoire divergence in the former pathogen is more significant to immune escape than in the latter, because *A. marginale* generates new variants from mosaic gene formation of fewer *msp2/p44* genes.

### 2.4. Tick Host Cell Genome

Miller and collaborators published a draft genome of an uninfected ISE6 cell line that indicated the presence of a contig with sequences matching an endosymbiont *Rickettsia* spp. [26]. We examined their draft sequence (GCA_002892825.2) further, as the presence of such sequences could potentially cause mis-assemblies of *A. phagocytophilum* genomes from infected ISE6 cultures. Contig PKSA02002878.1 contained numerous representatives of rickettsial conjugal transfer genes. To find evidence of this endosymbiont in our infected ISE6 cultures, we used the ApSheep_NorV1 Illumina and PacBio sequence assemblies to identify contigs matching PKSA02002878.1 of Miller et al. This analysis revealed a contig of size 37,330 base pairs containing conjugative *tra*-like genes sequences potentially derived from the *Ixodes scapularis* endosymbiont *Rickettsia buchneri* [27,28]. This contig was 99% identical with PKSA02002878.1 from position 593820-631150. We did not find evidence for the presence of a complete endosymbiont genome suggesting that these rickettsial conjugation transfer gene sequences had been incorporated into the ISE6 genome (Figure 8A). PCR amplicons corresponding to the ISE6 genome-*Rickettsia traA, traB*, *traD*, *traG*, and *traH* junctions were detected in genomic DNA extracted from both uninfected ISE6 cells and ISE6 cells infected with ApSheep_NorV1 *A. phagocytophilum*, confirming the presence of Rickettsia-*tra* gene sequences in the ISE6 cell line (Figure 8B). The obtained *tra* nucleotide sequences shared 100% identity with homologous sequences from *R. parkeri* (GenBank accession # CP040325.1), *R. bellii* isolate An04 (GenBank accession # CP015010.1), *R. rhipicephali* strain HJ#5 (GenBank accession # CP013133.1), and *R. philipii* str. 364D (GenBank accession # CP003308.1). Additional PCR analysis using genomic DNA extracted from different tick cell lines indicated that these sequences are only found in ISE6 cells and are maintained during continuous cell passages (Table 2, Appendix A). The *Rickettsia*-genus specific 17kDa antigen was not detected, providing additional evidence that a complete genome from a *Rickettsia* endosymbiont is not present in the ISE6 cell line (Table 2). Rickettsia-*tra* amplicons were not detected in the negative control sample that corresponds to genomic DNA from *Aedes vexans* mosquito-embryo cells (AVE1) [29] (Table 2). Moreover, although present in the uninfected ISE6 cells, *tra* sequences were not present in the assembled *A. phagocytophilum* genome of ApSheep_NorV1. There is a growing realization of the acquisition of foreign DNA by invertebrate genomes [30]. Horizontal gene transfer of sequences between bacterial endosymbionts and insects and nematodes is particularly widespread. *Wolbachia* DNA integration events in arthropod genomes have been described ranging from insertion of small fragments of < 500 base pairs to nearly complete genomes [31]. *Rickettsia* strains commonly harbor conjugation genes either on plasmids or within the bacterial genome and are transmitted horizontally between strains [32]. Interestingly, the frequent horizontal transmission of these conjugation genes results in the decoupling of Rickettsial phylogenies derived from conjugation genes compared to other methods such as multi-locus strain typing (MLST).

## 3. Conclusions

Analysis of these 28 complete or draft genomes of *A. phagocytophilum* revealed a surprising amount of strain diversity, in some cases approaching the proposed limits for species separation, although overall synteny was maintained. Much of the observed diversity was in the large repertoire of *msp2/p44* genes that encodes the immunodominant variable surface antigen Msp2/p44. 

We showed that the Norwegian field strain ApSheep_NorV1 is a mixed infection of two different *A. phagocytophilum* genotypes. Based on our analysis, it is possible that maintenance of these two different genotypes in the immunocompetent host is also due to differences in their *msp2/p44* repertoires. This has implications for vaccination as the immune responses against a genotype carrying a different *msp2/p44* repertoire to the vaccine may be less effective.

The presence of conjugational *tra* genes shown here in uninfected ISE6 cell lines suggests that sharing of these sequences is not limited to the bacteria but may occur with the host genome. It is not known whether these *tra* sequences are expressed in the host tick cell, but this warrants further investigation as does any possible effect of these sequences on infections of the cell lines.

## 4. Materials and Methods

### 4.1. Anaplasma Phagocytophilum Genome Sequences

The complete or draft genomes from 28 strains of *A. phagocytophilum* used for comparative analysis were: *Anaplasma phagocytophilum* str. HZ2 (GenBank accession # CP006616), *Anaplasma phagocytophilum* str. NYW (GenBank accession # LAOG00000000), *Anaplasma phagocytophilum* str. NCH-1 (GenBank accession # LANT00000000), *Anaplasma phagocytophilum* str. Webster (GenBank accession # LANS00000000), *Anaplasma phagocytophilum* str. WI1 (GenBank accession # LAOF00000000), *Anaplasma phagocytophilum* str. HGE1 (GenBank accession # APHH00000000), *Anaplasma phagocytophilum* str. HGE2 (GenBank accession # LAOE00000000), *Anaplasma phagocytophilum* str. Dog 2 (GeneBank accession # CP006618), *Anaplasma phagocytophilum* str. JM (GeneBank accession # CP006617), *Anaplasma phagocytophilum* str. CR1007 (GeneBank accession # LASO00000000), *Anaplasma phagocytophilum* str. Annie (GeneBank accession # LAON00000000), *Anaplasma phagocytophilum* str. MRK (GeneBank accession # JFBH00000000), *Anaplasma phagocytophilum* str. CRT35 (GeneBank accession # JFBI00000000), *Anaplasma phagocytophilum* str. CRT38 (GeneBank accession # APHI00000000), *Anaplasma phagocytophilum* str. CRT53 (GeneBank accession # LAOD00000000), *Anaplasma phagocytophilum* str. ApMUCO9 (GeneBank accession # LANV01000000), *Anaplasma phagocytophilum* str. ApNP (GeneBank accession # LANW01000000), *Anaplasma phagocytophilum* str. BOV-1_179 (GeneBank accession # CCXQ00000000), *Anaplasma phagocytophilum* str. Cow1 (GenBank accession # FLLR00000000), *Anaplasma phagocytophilum* str. Cow 2 (GenBank accession # FLMA00000000), *Anaplasma phagocytophilum* str. Cow 3 (GenBank accession # FLMB00000000), *Anaplasma phagocytophilum* str. Cow 4 (GenBank accession # FLLZ00000000), *Anaplasma phagocytophilum* str. Cow 5 (GenBank accession # FLMD00000000), *Anaplasma phagocytophilum* str. RoeDeer (GenBank accession # FLME00000000), *Anaplasma phagocytophilum* str. Horse 1 (GenBank accession # FLMF00000000), *Anaplasma phagocytophilum* str. Horse 2 (GenBank accession # FLMC00000000), *Anaplasma phagocytophilum* str. Norway Variant 1 (GenBank accession # CP046639), and *Anaplasma phagocytophilum* str. Norway Variant 2 (GenBank accession # CP015376).

### 4.2. Maintenance of Uninfected Tick Cell Lines and Cultivation of ApNorV1 Mutants

For this work, we used 11 tick cell lines derived from embryonic stages from the following tick species: *Ixodes scapularis* ISE6, ISE18, ISE24, IDE2, and IDE8 [36], *Ixodes ricinus* (IRE11) [37], *Amblyomma americanum* (AAE2) [38], and *Rhipicephalus microplus* (BME26) [39]. Uninfected tick cell cultures were maintained in modified Leibobitz’s L15 medium prepared as described previously [36,37,38,40,41] and supplemented with 5% FBS, 5% tryptose phosphate broth (TPB; Difco, Becton, Dickinson, Baltimore, MD, USA), and 0.1% bovine lipoprotein concentrate (LPC; MP-Biomedical, Irvine, CA, USA). The ISE6 cells infected with ApSheep_NorV1 mutants (CL1A2, CL2B5, and CL3D3) [2] were maintained in modified Leibobitz’s L15 medium, as above, but additionally supplemented with 0.25% NaHCO_3_, 25 mM HEPES buffer, and 100 µg/mL of spectinomycin and streptomycin (Teknova, Hollister, CA, USA) [41].

### 4.3. Preparation of A. phagocytophilum from Infected Sheep

Lambs were raised in an indoor environment with barriers against tick entry and infestation. They were intravenously infected with a cryopreserved blood stabilate of the 16S genotype of *A. phagocytophilum* GenBank M73220 (ApSheep_NorV1), as described previously [2]. Infections were monitored by microscopy and blood harvested at maximum bacteremia. Buffy coats were prepared and washed as described [42] and stored at −80 °C.

### 4.4. DNA Extraction and PCR Analysis

Genomic DNA was prepared from the frozen pellets of infected sheep buffy coat or ISE6 cells and sequenced on either Roche/454, Illumina, or Pacific Biosciences platforms [23,42]. Sequence assemblies utilized Pacific Biosciences SMRT Pipe [43], Canu [44], and Spades [45] assembly software. For draft genome sequences available in GenBank, contigs were ordered prior to alignment with Mauve [46], using as reference a finished genome with similar average nucleotide identity as determined by JSpecies software [18].

Frozen whole blood from two lambs (#4203 and #4210) experimentally infected with the Norwegian field strain ApSheep_NorV1, collected at different time points (days 8, 22, 50, 69, and 93 post-inoculation) [10], was processed for DNA extraction using the Zymo Quick-gDNA Blood Microprep (Zymo, Irvine, CA, USA) following manufacturer’s instructions. Genomic DNA was extracted from 400 μl of a 5-ml culture of uninfected ISE6 cells (containing approximately 5 × 10^6^ cells), or cells infected with the *A. phagocytophilum* mutants CL1A2, CL2B5, or CL3D3 [2] using the Quick gDNA microprep kit (Zymo, Irvine, CA, USA) following manufacturer’s instructions. DNA concentration from each sample was determined using the Qubit dsDNA HS assay kit (ThermoFisher scientific, Waltham, MA, USA) on a Qubit fluorometer (ThermoFisher scientific, Waltham, MA, USA).

Two sets of primers targeting a region within the virB6-2 gene were designed to differentiate between the major and minor variants (Table 3). Ten ng of DNA extracted from *A. phagocytophilum*-infected blood or cell cultures were used as template in 50 µl reactions that contained 200 μM dNTPs, 0.25 μM of forward and reverse primers, and 1.25 units of PrimeSTAR GXL DNA polymerase (Takara, San Jose, CA, USA). PCR amplification was performed in a BioRad PTC-100 thermal cycler (BioRad, Hercules, CA, USA) with the following conditions: 94 °C 2 min, followed by 35 cycles at 98 °C for 10 s, 55 °C for 15 s, and 68 °C for 1 min, and a final extension cycle at 68 °C for 5 min. PCR products were separated electrophoretically using a 2.5% Seakem LE (Lonza, Morristown, NJ, USA) agarose gel, and stained with SYBR Gold nucleic acid gel stain (ThermoFisher scientific, Waltham, MA, USA) for visualization.

### 4.5. Determination of msp2/p44 Repertoires in Each Strain Genome of A. phagocytophilum

Previous alignments of *msp2/p44* genes have identified two conserved regions flanking a hypervariable region [22]. In the 5’-conserved region, we identified the 11 base sequence aaggagttagc that was also shared between *msp2/p44* genes of *A. phagocytophilum* and *msp2* genes of *Anaplasma marginale*. The linux "grep" command was used to extract all instances of this sequence plus the downstream 469 bases from all 28 complete and draft genome sequences of *A. phagocytophilum* in both forward and reverse orientations. Then, an all-against-all blastn search was conducted with the extracted *msp2/p44* genes using an e value of 0.001 to generate tabular delimited output for importation into Excel spreadsheets. As the Blastn output was 1,310,704 rows total, the data was separated into two files for importation into spreadsheets with the linux “split” command. Excel filters were then applied to identify the *msp2/p44* sequences in each genome with at least 99% identity over at least 456 bases. A further filter was applied to verify that matching genes encoded at least one of these known conserved Msp2/p44 amino acid sequence characteristics [10]: N-terminal KELAY and N- or C-terminal LAKT. The above methods will clearly be most accurate in determining the repertoires from the finished rather than draft genomes.

### 4.6. Identification of Symbiotic Rickettsia tra Genes in ISE6 Cells 

Genomic DNA from ISE6 cells, either uninfected or infected with *A. phagocytophilum* ApSheep_NorV1mutants, was obtained as described above. Fifty ng of genomic DNA was used as template in 50 µL reactions that contained 200 μM dNTPs, 0.25 μM of primers targeting rickettsial *traA*, *traB*, *traD*, *traG*, and *traH* genes (Table 3), and 1.25 units of PrimeSTAR GXL DNA polymerase (Takara). PCR amplification was performed in a BioRad PTC-100 thermal cycler (BioRad) with the following conditions: 94 °C 1 min, followed by 30 cycles at 94 °C for 30 s, 60 °C for 30 s, and 72 °C for 1 min, and a final extension cycle at 72 °C for 5 min. PCR products were separated electrophoretically using a 1% Seakem LE (Lonza, Morristown, NJ, USA) agarose gel, and stained with SYBR Gold nucleic acid gel stain (ThermoFisher scientific, Waltham, MA, USA) for visualization.

### 4.7. Pan Genome Analysis

The 28 *A. phagocytophilum* genomes were annotated with Prokka [47] and the obtained gff3 files used as input for Pangenome inference using Roary [48] (95% BLAST v2.6.0 percent identity cut-off) that clusters full length genes into core (hard and soft core) and accessory (shell and cloud) genomes. Hard- and soft-core genes are found in >99% and 95–99% of the genomes, respectively. The shell and cloud genes (accessory) are found in 15–95% and < 15% of the genomes, respectively. This analysis produced a presence/absence of core and accessory genes matrix and a phylogenetic tree. The phylogenetic tree was inferred from a multi-FASTA alignment of core genes using FastTree 2 version 2.1.9 [49]. The core-genome tree was then compared with the presence/absence of core and accessory genes matrix and visualized with the interactive viewer Phandango [50].

## Figures and Tables

**Figure 1 pathogens-11-00601-f001:**
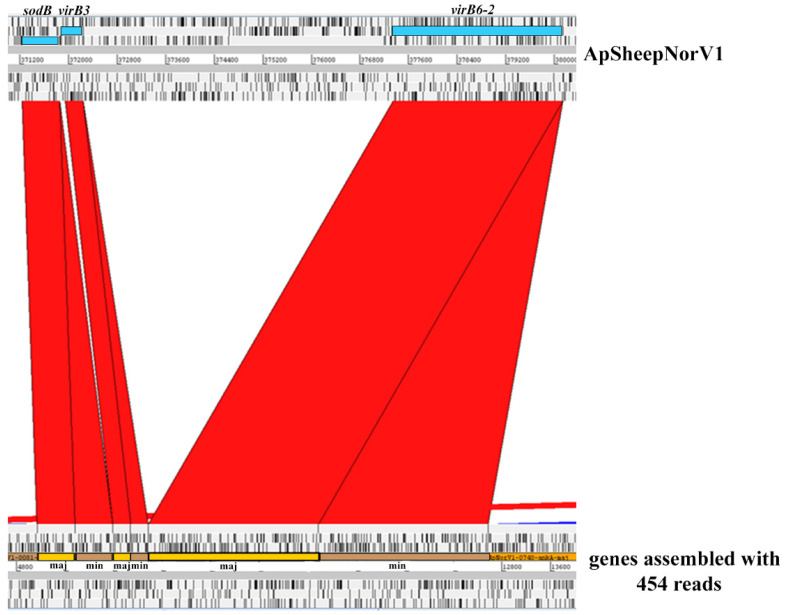
A comparison between the *sodB*, *virB3*, and *virB6-2* genes (blue) from the ApSheep_NorV1 genome (CP046639) and *sodB*, *virB3*, and *virB6-2* sequences obtained from the 454 reads. The ApSheep_NorV1 genome was obtained from organisms in tick culture whereas the 454 reads were obtained from an acute blood-stage infection. Yellow and brown boxes correspond to heterozygous copies of *sodB*, *virB3*, and *virB6-2* genes indicative of a mixed infection with the major and minor genotypes, respectively. Genes from both panels are translated into six reading frames with black bars indicating stop codons, using the Artemis Comparison Tool (Sanger Institute). Bands in red represent homology between the *sodB*, *virB3*, and *virB6-2* sequences from the newly finished ApSheep_NorV1 genome and assembled sequences from the 454 reads.

**Figure 2 pathogens-11-00601-f002:**
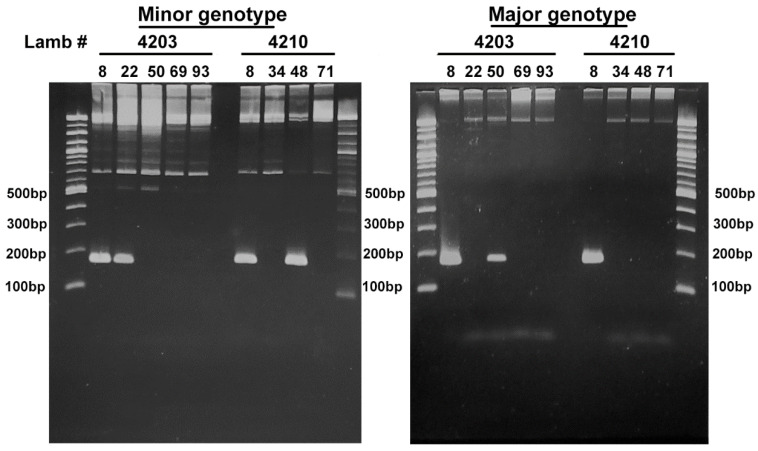
Agarose gel electrophoresis showing the detection of ApSheepNorV1 major and minor genetic variants during consecutive rickettsemia peaks. Sampling times for lamb #4203 were days 8, 22, 50, 69, and 93 post-infection or 8, 34, 48, and 71 post-infection for lamb #4210.

**Figure 3 pathogens-11-00601-f003:**
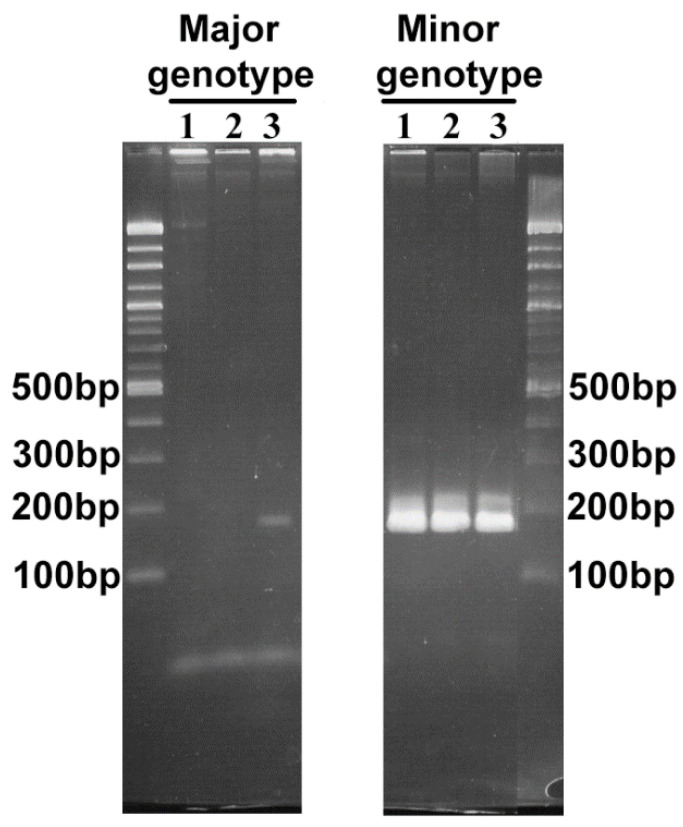
Agarose gel electrophoresis of PCR products using primers targeting the *virB6-2* gene from ApSheepNorV1 major (169 bp) and minor (167bp) variants. DNA amplicons from ISE6 infected with 1. CL1A2, 2. CL2B5, and 3. CL3D3 mutants.

**Figure 4 pathogens-11-00601-f004:**
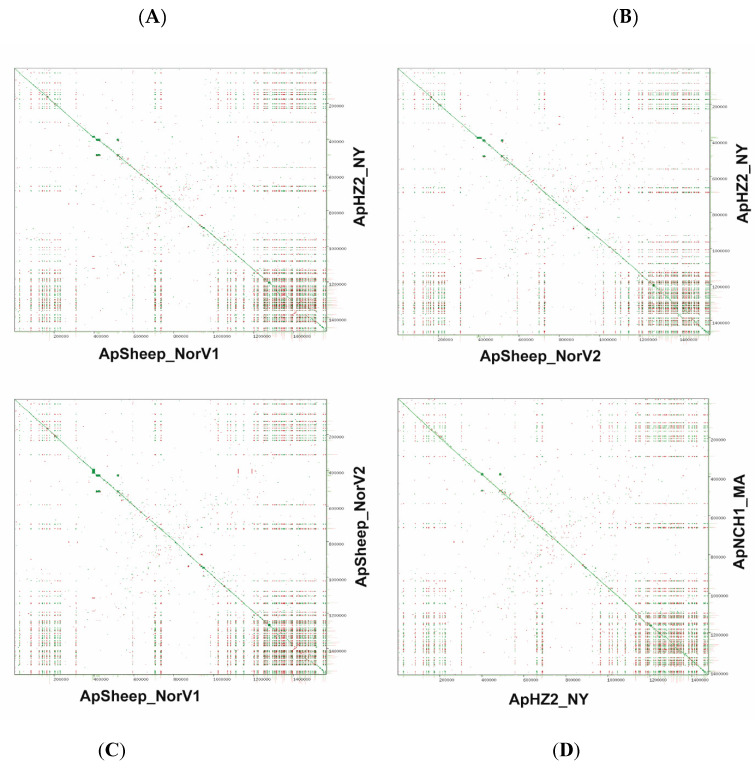
Homology and synteny analysis using YASS genomic dot-plots. Dot-plot comparisons of (**A**) ApSheep_NorV1 vs. ApHZ2_NY, (**B**) ApSheep_NorV2 vs. ApHZ2_NY, (**C**) ApSheep_NorV1 vs. ApSheep_NorV2, and (**D**) ApHZ2_NY vs. ApNCH1_MA.

**Figure 5 pathogens-11-00601-f005:**
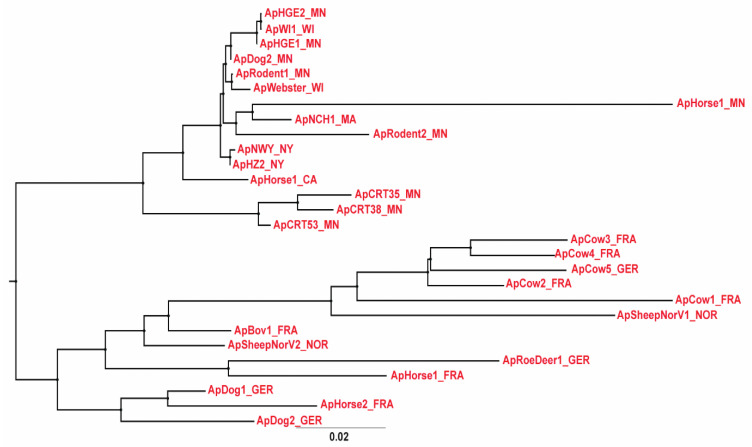
Taxonomic relationships of *A. phagocytophilum* strains. Genome sequences were aligned with MAFFT and trees generated following analysis of substitution models with Jmodeltest2.

**Figure 6 pathogens-11-00601-f006:**
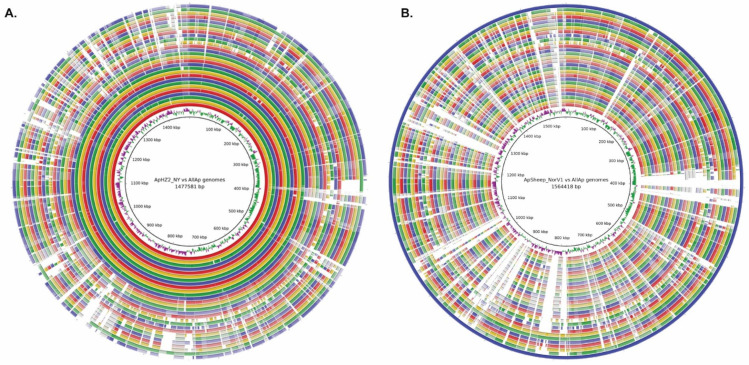
Differences across the genomes of 28 *A. phagocytophilum* strains. Genome comparisons used Blastn and the Blast Ring Image Generator. The image shows the similarity between either the HZ2_NY reference strain (**A**) or the ApSheep_NorV1 reference strain (**B**) and the other sequences as concentric rings. Key: From center, Ring 1, base numbering; Ring 2, GC skew; Rings 3–9, U.S. strains from humans in NY, MA, WI, and MN; Rings 10–13, U.S. strains from rodents, dogs, and horses in MN; Ring 14, U.S. strain from horse in CA; Rings 15–17, U.S. CRT (Ap-variant 1) strains; Rings 18–21, European strains from dogs and horses; Rings 22–28, European strains from roe deer and cattle, Rings 29–30, strains from Norwegian sheep. Regions of sequence difference in the range 100–92% identity are shown as a fading color gradient and < 92% identity as gaps.

**Figure 7 pathogens-11-00601-f007:**
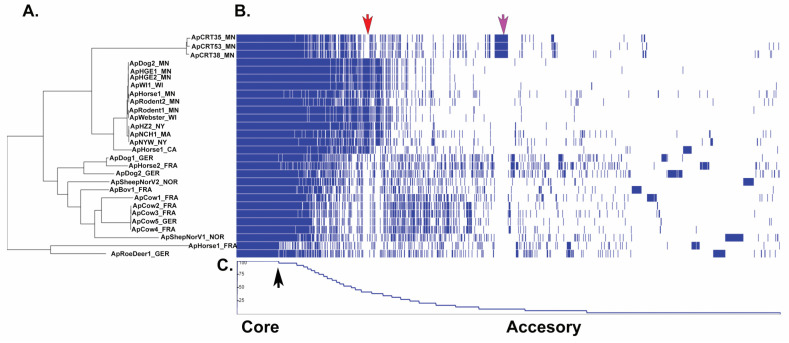
Distribution pattern of genes in the *A. phagocytophilum* Pan-genome. (**A**) Maximum likelihood phylogenetic tree based on the *A. phagocytophilum* core genome. (**B**) Heatmap showing gene presence (blue) or absence (white) in each of the 28 *A. phagocytophilum* strains. (**C**) Plot showing the gene frequency across the *A. phagocytophilum* pan-genome where genes left of the black arrow correspond to the core genome and genes to the right of the arrow correspond to accessory genes (present or absent). Accessory genes unique to Ap-ha related strains (red arrow) and accessory genes unique to CRT strains (Ap-variant 1) (magenta arrow) are indicated.

**Figure 8 pathogens-11-00601-f008:**
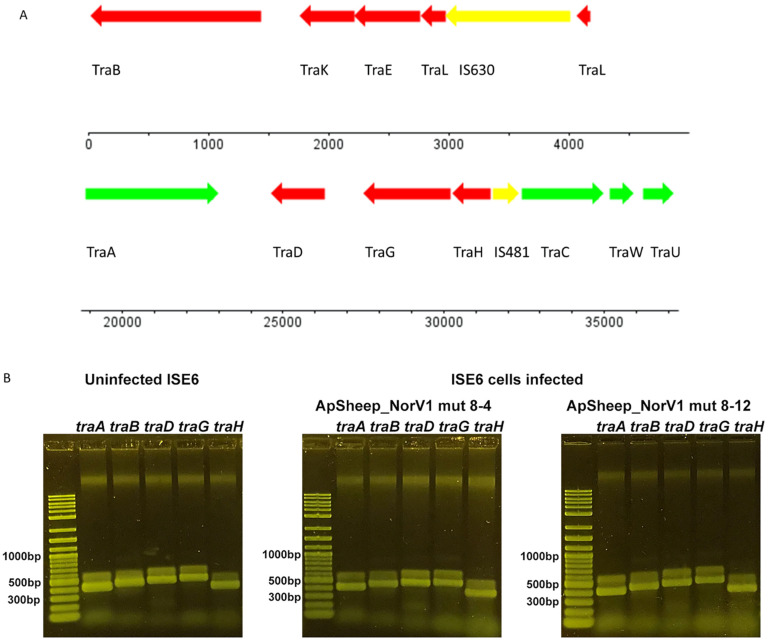
Rickettsia Tra genes present in both uninfected and *A. phagocytophilum*-infected ISE6 tick cells. (**A**) Contig of 37,330 base pairs present in ISE6 cells infected with ApSheep_NorV1. Tra-like sequences and IS630 or IS481 family transposases are indicated; (**B**) Amplification by PCR of Tra sequences from either uninfected or *A. phagocytophilum*-infected ISE6 tick cells. Primer sequences and amplicon sizes are described in Table 3.

**Table 1 pathogens-11-00601-t001:** Proportions of sequencing reads for two genes representing major and minor population types of *A. phagocytophilum* isolated from Norwegian sheep.

Gene	Conserved Flanking Sequences	Major Read Type/Total Reads	Minor Read Type/Total Reads
*omp1X*	CTGGACGTGTTCTCTGCT……GGGAAGTATAATCCTTCA	31/36	5/36
*omp1X*	GTAAAGCATAATATAGCG……CAGAGTATCAGCACTGAT	30/32	2/32
*virB6-2*	CACATAAGTGGTGGTGGT……TTACTAAGAAAAGAGGAG	17/22	5/22
*virB6-2*	ACACTTGTGCCTAAATCG……ATTGCAGGCAGTTACACT	17/23	2//34
Total		17/24	14/124
Percentage		17/25	11%

**Table 2 pathogens-11-00601-t002:** Presence/absence of rickettsial *tra* genes in different uninfected tick cell lines.

Tick Cell Line and Passage # (p.)	(a)Tick F/R	(b)17kDa	traA	traB	traD	traG	traH
ISE6 p.28	+	−	+	+	+	+	+
ISE6 p. 133A	+	−	+	+	+	+	+
ISE6 p. 133B	+	−	+	+	+	+	+
ISE18 p. 43	+	−	−	−	−	−	−
ISE24 p. 6 A	+	−	−	−	−	−	−
ISE24 p.6 B	+	−	−	−	−	−	−
IDE2 p. 61	+	−	−	−	−	−	−
IDE8 p. 36 A,B	+	−	−	−	−	−	−
IRE11 p. 78 A,B	+	−	−	−	−	−	−
AAE2 p. 14	+	−	−	−	−	−	−
BME26 p. 89	+	−	−	−	−	−	−
AVE1 p. 9	+	−	−	−	−	−	−

(+) PCR positive; (−) PCR negative; (a)Forward and Reverse Primers (F/R) targeting tick mitochondrial 16S rRNA or β-actin genes; (b)PCR targeting Rickettsia-genus specific 17kDa antigen gene.

**Table 3 pathogens-11-00601-t003:** Oligonucleotides used in this study.

Primers Targeting *A. phagocytophilum* Variants			
	Sequence (5’ to 3’)	Amplicon size (bp)	Target
AB1751	GGTGCTCTTAAAAAACCTAGCGC	169	Major variant
AB1752	CAGCCGCAAAGTTCTTTACTCTAT		
AB1753	GGGCTTTTAAAGCATAGCTT	167	Minor variant
AB1754	TAGCTGCAGGATCCTTTACTCCCC		
**Primers using for *Rickettsia*-*tra* genes in ISE6 cells analysis**			
AB2111	ATGAAGCAGGGATGGTAGGT	415	*tra*A
AB2112	AGTAACTCCCTGATGCCTTGA		
AB2113	TTCCAACCACCACACCAGTA	450	*tra*B
AB2114	AGGTCAAAACTTCCCGAGGT		
AB2115	AACGCAGCCACCATATTTCC	473	*tra*D
AB2116	AGTTGGTACTCGCTGAAGGA		
AB2117	TGGTGAGTACTTGCCCACAT	491	*tra*G
AB2118	GCCTTGCAGCAGATGAATTC		
AB2119	AATTGAGGCAGATTCTCCGC	383	*tra*H
AB2120	AGGGCGATTGTGAAATGACT		
T1 forward	CCGGTCTGAACTCAGATCAAGT	460	Mitochondrial 16S rRNA [33]
T1 reverse	CTGCTCAATGATTTTTTAAATTGCTGTGG		
T2 forward	GGTATCGTGCTCGACTC	450	Tick β-actin [34]
T2 reverse	ATCAGGTAGTCGGTCAGG		
Rr17.61p	GCTCTTGCAACTTCTATGTT	434	17kDa [35]
Rr17.492n	CATTGTTCGTCAGGTTGGCG		

## Data Availability

Data supporting the reported results are available here in Appendix A.

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
