# Peer review of "Comparative Whole Genome Analysis of an Anaplasma phagocytophilum Strain Isolated from Norwegian Sheep"

_pathogens, 2022, doi:10.3390/pathogens11050601_

Round 1

Reviewer 1 Report

The review report is attached

Author Response

Pease see the attachment

Reviewer 2 Report

Anaplasma phagocytophilum is one of the most important tick-borne pathogens, having zoonotic potential. The articles that approach this pathogen are attractive and exciting, the present manuscript framing in this line. However, it has some shortcomings.

Abstract section.

  • Lines 18-20: I propose: "Although A. phagocytophilum is characterized by increased genetic diversity and complex infections (co-infection and superinfection) in ruminants and other animals are common, however, the underlying genetic basis of intra-species interactions and host-specificity remains unexplored.". It seems to be more understandable!
  • Line 27: isn't it better "...we conclude" instead of "we propose"? It is more accurate to state: "we conclude that complex infections can be attributed to infection with strains..." than to propose that "that complex infection can be attributed". "We propose" doesn't match with "can be attributed".

General comments

After reading the Abstract, I tried to review each section and subsection separately but failed. I read your manuscript three times and had to check if it is a research article or a review, according to MDPI's Types of Publications. So, it is an Article that should present original research.

Your research is complex, of great interest, and based on modern techniques, but the way you convey information to readers is complicated and difficult to understand. I will explain why.

The sections of a Research Manuscript, according to MDPI, are the Introduction, Results, Discussion, Materials and Methods, and optional, Conclusions.

You have chosen to combine the Results and Discussion sections. In each subsection of this combined section, you made a short introduction (bibliographic sentences), after which you mixed the actual results with the discussions, also presenting some elements related to Materials and Methods. The result was an intermixture challenging to follow and understand. The Result section should provide a concise and precise description of the experimental results and their interpretation, and further, you will draw conclusions based on those results. This doesn't happen in your intermixed manuscript.

Regarding the Conclusions section, your manuscript is one of the very few articles (among the countless) reviewed by me in which the conclusions contain bibliographical references. The conclusions represent the finality of a study, are stated by the authors, and should not be referenced!

I propose a reorganization of the manuscript. Separate the results and the discussions! Present the results clearly and concisely, and in the Discussion section, interpret them and integrate them into the results of research conducted around the world. This is how your manuscript will become more readable and understandable!

Author Response

Pease see the attachment

Reviewer 3 Report

The study was well conducted with rigour and with adapted tools.

Only minor revisions are requested :

Lines 205 -208 : clarify this two sentences

Line 264 : is the meaning of « F/R » Forward/Reverse ?

Line 371 : Ixodes ricinus

Table 1 : « conserved flanquing sequence »

Reviewer 4 Report

This menu is suitable for Pathogens.

Author Response

Reviewer #4 Have not particular revisions to our manuscript

Round 2

Reviewer 2 Report

You revised the Conclusions and removed the bibliographic references that were extremely disturbing to me. You have slightly modified the Results and Discussions, improving the quality of your manuscript. As such, I consider it to be publishable in its current form. Congratulations!